# Divergent Immune Pathways in Coronary Artery Disease and Aortic Stenosis: The Role of Chronic Inflammation and Senescence

**DOI:** 10.3390/ijms26115248

**Published:** 2025-05-29

**Authors:** José Joaquín Domínguez-del-Castillo, Pablo Álvarez-Heredia, Irene Reina-Alfonso, Maria-Isabel Vallejo-Bermúdez, Rosalía López-Romero, Jose Antonio Moreno-Moreno, Lucía Bilbao-Carrasco, Javier Moya-Gonzalez, María Muñoz-Calero, Raquel Tarazona, Rafael Solana, Alexander Batista-Duharte, Ignacio Muñoz, Alejandra Pera

**Affiliations:** 1Cardiovascular Pathology Group (GA09), Maimonides Biomedical Research Institute, University of Cordoba, Reina Sofia University Hospital, Avda Menedez Pidal s/n, 14004 Cordoba, Spain; ep2docaj@uco.es (J.J.D.-d.-C.); rosalia.lopez.romero.sspa@juntadeandalucia.es (R.L.-R.); josea.moreno.sspa@juntadeandalucia.es (J.A.M.-M.); lucia.bilbao.sspa@juntadeandalucia.es (L.B.-C.); javier.moya.sspa@juntadeandalucia.es (J.M.-G.); maria.munoz.calero.sspa@juntadeandalucia.es (M.M.-C.); ignacio.munoz.carvajal.sspa@juntadeandalucia.es (I.M.); 2Department of Cell Biology, Physiology and Immunology, University of Cordoba, Avda Menedez Pidal s/n, 14004 Cordoba, Spain; rsolana@uco.es; 3Immunology and Allergy Group (GC01), Maimonides Biomedical Research Institute of Cordoba (IMIBIC), University of Cordoba, Reina Sofia University Hospital, Avda Menedez Pidal s/n, 14004 Cordoba, Spain; pablo.alvarez@imibic.org (P.Á.-H.); z02reale@uco.es (I.R.-A.); isabel.vallejo@imibic.org (M.-I.V.-B.); 4Hospital Nostra Senyora de Meritxell, Avinguda Fiter i Rossell 13, AD700 Escaldes-Engordany, Andorra; 5Immunology Unit, Department of Physiology, University of Extremadura, 10003 Cáceres, Spain; rtarazon@unex.es

**Keywords:** atherosclerosis, coronary artery disease, aortic stenosis, immune profiling, flow cytometry, immunosenescence, machine learning, predictive model

## Abstract

Coronary artery disease (CAD) remains a major cause of cardiovascular morbidity and mortality, with growing evidence linking immune dysregulation to its pathogenesis. Aortic stenosis often coexists with CAD (ASCAD), representing an advanced disease form. This study investigates immune pathways in isolated CAD (iCAD) and ASCAD. For this purpose, peripheral blood from 72 individuals (healthy donors, iCAD, and ASCAD patients) was analysed via flow cytometry to assess immune populations. Circulating cytokine levels were measured, and machine learning models identified predictive immune biomarkers. Our data showed that both iCAD and ASCAD patients exhibited immune dysregulation, with reduced dendritic cells, basophils, NK cells, B cells, and T cells, alongside lower frequencies of DCs, lymphocytes, CD8+CD28+ T cells, and CD57+ T cells. Elevated IL-15 and fractalkine, but reduced IL-8 and MCP-1, suggest impaired monocyte and neutrophil mobilisation due to immune cell sequestration in vascular lesions. Distinct immune features emerged between iCAD and ASCAD. iCAD patients showed heightened immune activation, with increased inflammatory CD14+CD16+ monocytes, higher Treg frequencies, and greater CD4+ T cell differentiation into TEM and TEMRA phenotypes. In contrast, ASCAD patients exhibited pronounced immunosenescence, with higher neutrophil counts, lymphopenia, and increased NK and T cell cytotoxicity. Our predictive model distinguished iCAD from ASCAD with high accuracy, identifying CD4+ T cell memory subsets and CD57 expression as key discriminators. This study reveals iCAD as being driven by immune activation and ASCAD by immunosenescence and cytotoxicity. These insights advance CAD immunopathology understanding and support immune-based classification, particularly for ASCAD, where treatment remains limited to surgical intervention.

## 1. Introduction

Coronary artery disease (CAD) is a chronic condition characterised by the narrowing and blockage of coronary arteries due to the buildup of atherosclerotic plaques. These plaques originate from the accumulation of cholesterol, fatty substances, cellular waste products, calcium, and fibrin. Plaques can reduce or completely obstruct the blood flow to the heart muscle, leading to ischemia and potentially resulting in acute coronary syndrome and myocardial infarction [1].

CAD is a leading cause of morbidity and mortality worldwide, significantly contributing to the global burden of cardiovascular diseases. According to recent estimates, CAD is responsible for approximately 9 million deaths each year, accounting for nearly one-third of all global deaths. The prevalence of CAD continues to rise in low- and middle-income countries due to the increasing incidence of risk factors such as hypertension, diabetes, dyslipidaemia, smoking, and obesity. The burden of CAD extends beyond mortality, leading to significant disability and economic costs associated with its long-term treatment and management [2].

Recent research has recognised the crucial role of the immune system in the pathogenesis of CAD with atherosclerosis understood as a chronic inflammatory disease resulting from immune system alterations. Arterial wall inflammation is a key factor in plaque formation, progression, and destabilisation. The immune response in atherosclerosis involves both the innate and adaptive immune systems, with various immune cells, including macrophages, T cells, and B cells, contributing to the inflammatory processes underlying CAD and other atherosclerotic-based diseases such as aortic stenosis (AS) (Figure 1) [3,4].

Immunosenescence, the gradual decline in immune function associated with ageing, has significant implications for CAD. As the immune system ages, there is a shift in the balance of immune cell populations, characterised by a decrease in naïve T cells and an increase in memory and senescent T cells [5], characterised by the loss of CD28 and gain of CD57. These cells are terminally differentiated, highly proinflammatory, and cytotoxic and are resistant to apoptosis and to the control of regulatory T cells [6,7,8,9]. It has been shown that CD28^null^CD57+ T cells can directly and indirectly kill endothelial cells [10], having the necessary arsenal to cause vascular damage. Certainly, the expansion of CD28^null^CD57+ T cells has been linked to a heightened inflammatory response and increased oxidative stress, both of which contribute to the pathogenesis of atherosclerosis and CAD [11,12]. Thus, this T cell phenotype shift as well as other changes occurring during immunosenescence contribute to the establishment of a chronic pro-inflammatory state, which ultimately can lead to the development and progression of atherosclerosis [13].

Furthermore, a study conducted by Moro-García et al. showed that patients with acute coronary syndrome (ACS) exhibited increased levels of highly differentiated, senescent T cells and reduced naive T cells, indicating an aged immune profile that correlated with worse clinical indicators, such as elevated SYNTAX scores and lower ejection fractions. Additionally, lipid profile parameters, such as high LDL levels, were also associated with increased T cell differentiation [14]. These data support the notion that immunosenescence might play a role in exacerbating CAD by impairing adaptive immune responses and underscore the utility of flow cytometry in evaluating immune dysregulation in these patients, potentially guiding future interventions aimed at modulating immunosenescence to mitigate disease progression.

CAD has been demonstrated to frequently occur in conjunction with aortic stenosis (AS) [15]. Calcific AS (CAS) is the first cause of valvular heart disease in high-income countries [16]. It refers to the progressive thickening and calcification of aortic valve leaflets. Studies show that up to 50% of patients with CAS have concomitant CAD. This strong association can be explained by the shared pathobiology and risk factors [17]. Many publications support the existence of a common pathophysiological mechanism in the form of atherosclerosis [17,18,19]. Despite this evidence, very limited literature has focused on identifying and characterising the common phenotypes and disparities in patients with isolated severe CAD and those with associated severe CAS.

Thus, detailed profiling of the changes occurring in immune cells can reveal specific changes, such as the accumulation of senescent T cells, which are associated with increased inflammation and vascular dysfunction. Understanding these cellular dynamics is crucial for developing targeted therapies that address the underlying immune mechanisms driving CAD, particularly in the ageing population. Thus, here we present a comprehensive flow cytometry analysis of the composition and phenotypes of peripheral immune cells from CAD patients, to accurately identify disease-specific immune signatures.

## 2. Results

### 2.1. Peripheral Immune Cell Profiles in CAD Patients

The primary focus of our study was to thoroughly characterise the immune profile of CAD patients (iCAD and ASCAD, Appendix A) to identify potential disease biomarkers. For this, we conducted a flow cytometry analysis of innate and adaptive immune cells (Appendix A), which revealed significant differences among patients and controls. These differences were observed in both the absolute numbers and relative proportions of multiple immune cell populations (Table 1, Table 2 and Appendix A), highlighting the alterations in both the overall immune cell counts and compositional balance across disease groups.

Compared to HD, CAD patients exhibited a higher total leukocyte count, although not statistically significant (*p* = 0.056). They also had an elevated neutrophil-to-lymphocyte ratio (NLR) due to increased neutrophil and decreased lymphocyte counts. The decrease in lymphocyte counts also translated into a reduced lymphocyte-to-monocyte ratio (LMR), despite conserving the numbers of monocytes. Additionally, CAD patients had lower numbers of basophils and dendritic cells (DCs, mDCs, and pDCs), and monocyte subsets CD14+CD16- and CD14^lo^CD16++. Furthermore, the overall reduction in lymphocyte counts was associated with decreases in B cell counts, total T cells (including, CD4+, CD8+, TCRγδ, CD4+CD8+, and Treg subsets), as well as total and CD56^dim^ NK cell counts (Table 1).

In addition to alterations in the absolute cell numbers, CAD patients also exhibited significant changes in the relative proportions of several immune cell populations. The percentage of neutrophils was increased, while lymphocyte proportions were reduced, consistent with the changes observed in those subset counts. Proportional reductions were also observed in dendritic cell subsets (total DCs, mDCs, and pDCs), CD14^lo^CD16++ monocytes, and basophils. Among the lymphocyte subsets, CAD patients showed lower percentages of B cells, T cells (including, CD4+, CD8+, TCRγδ, CD4+CD8+, and Treg), and CD56^dim^ NK cells. Conversely, the proportion of CD56^bright^ NK cells was elevated, reflecting a shift in the NK cell subpopulation balance (Table 2).

We further investigated NK and T cell phenotypes, revealing a pattern suggestive of accelerated immunosenescence in CAD patients. Specifically, in terms of absolute numbers, CAD patients exhibited a decline in CD56^dim^ NK cells, although these cells expressed higher levels of CD4 and CD57, suggesting enhanced cytotoxic potential (Appendix A). Additionally, we observed a decline in TN, TCM, and TEM CD4+ T cell counts, compared with HD. Contrarily, CD4+ HLA-DR+ TEMRA cell counts were elevated, indicating increased immune activation (Appendix A). Although the absolute numbers of CD8+ T cell memory subsets remained largely similar between patients and controls, CAD patients showed a notable accumulation of CD57+ CD8+ T cells and a significant reduction in CD8+CD127+ cells (Appendix A).

Proportional analysis of these populations mirrored the absolute numbers (Figure 2, Appendix A). CAD patients showed an increased percentage of TEM and TEMRA CD4+ T cells, with a concurrent decrease in TN and TCM subsets, reflecting a shift toward a more differentiated immune profile. The percentage of CD4+CD127+ T cells was also lower, while CD4+ T cells expressing HLA-DR were more frequent, indicative of an activated phenotype (Figure 2A). CD8+ T cells from CAD patients exhibited a more proinflammatory and cytotoxic profile, with a higher proportion of CD57+ cells and a reduction in CD28+ cells. The frequency of CD8+CD127+ cells was also diminished (Figure 2B). Moreover, the proportion of overall CD56^bright^ and HLA-DR+ CD56^bright^ NK cells was elevated, while CD56^dim^ NK cells were proportionally reduced (Figure 2C). In addition to CD57+ T and NK cell accumulation, CAD patients displayed a higher per-cell expression of CD57 within CD4+, CD8+, and CD56^dim^ NK subsets, with CD4+ T cells exhibiting the highest levels (Figure 2D).

### 2.2. Peripheral Immune Cell Profiles in iCAD and ASCAD Patients

Given that our patient cohort included individuals with and without AS, we divided the patient group to investigate whether these alterations were primarily attributable to CAD independently of the contribution of AS. The analysis of absolute numbers showed that the changes observed regarding neutrophil and lymphocyte counts, the NLR and LMR, were attributable to the combination of CAD and AS, and did not occur in iCAD. Likewise, lymphopenia was more pronounced in ASCAD. Moreover, the NK cell counts were significantly elevated in ASCAD. In contrast, iCAD patients uniquely exhibited decreased absolute counts of CD14+CD16^lo^ and CD14^lo^CD16++ monocytes (Table 1, Figure 3).

Proportional analysis revealed that iCAD patients exhibited a higher frequency of CD14+CD16+ inflammatory monocytes and an increased proportion of CD56^bright^ NK cells and Treg cells. In contrast, ASCAD patients showed a significantly higher percentage of neutrophils and a more pronounced reduction in lymphocyte proportions compared to both HD and iCAD. Total NK cell frequencies were increased exclusively in ASCAD probably as an effect of the decline of both T and B cell proportions. There was also an increase in CD4^hi^CD8^lo^ T cells percentage only in ASCAD patients (Table 2).

The analysis of T and NK cell phenotypes, in terms of proportions, showed an association of AS with an increase in NK cells expressing CD57 and CD4 markers, while iCAD patients had increased frequencies of CD56^dim^ HLA-DR+ cells (Appendix A). Similarly, in T cells, the expression of CD57 was associated with the presence of AS, while in iCAD patients, T cells were more activated (HLA-DR+). Specifically, ASCAD patients had accumulations of T cells expressing the CD57 marker (including the memory subsets) (Appendix A). However, iCAD patients exhibited higher CD4+ and CD8+ T cell activation (HLA-DR+) and higher CD4+ T cell differentiation towards the TEM and TEMRA phenotypes (Appendix A).

In summary, compared to HD, iCAD patients were characterised by higher frequencies of CD14+CD16+ inflammatory monocytes, Treg cells, greater differentiation of CD4+ T cells (TEM and TEMRA), and general activation of NK and T cells and their subsets (Figure 2, Table 2 and Appendix A). In contrast, ASCAD patients were associated with a higher NLR and lower LMR (Table 1, Figure 3), higher proportions of NK cells, CD57+ NK cells, CD57+ CD56^dim^ NK cells, CD57+ and CD28^null^ T cells (CD4+, CD8+, and TCRγδ), and higher proportions of CD4+ and CD8 T cells expressing CX3CR1, although this last difference was not statistically significant (Appendix A).

When directly comparing iCAD and ASCAD patients’ immune cell counts, significant differences were found. In particular, ASCAD patients exhibited a significantly higher NLR and lower LMR than both controls and iCAD patients (Figure 3). Additionally, although both groups of patients had lower NK cell counts than the controls, iCAD patients had the lowest counts (Table 1). Furthermore, Treg counts, although generally reduced in patients, were significantly lower in ASCAD patients compared to iCAD patients (Table 1).

The differences in subset proportions among iCAD and ASCAD patients included increased percentages of neutrophils, CD14+CD16+ monocytes, CD56^bright^ NK cells, and Treg in iCAD patients (Table 2 and Appendix A). Regarding T and NK cell phenotypes, iCAD patients exhibited a more pronounced activation profile, while ASCAD patients showed an accumulation of cytotoxic NK and T cells (Appendix A).

Finally, to investigate whether the observed changes in T and NK cells translated into differences in their phenotypic profiles, we used the SPICE permutation test. This analysis showed significant differences in the phenotypic profiles of T and NK cells between CAD patients and HD, independently of AS (Figure 4). When comparing iCAD and ASCAD, we found a nonsignificant trend in CD4+ T cells’ profile, suggesting that these cells might play a distinct role in the progression of these conditions.

### 2.3. Inflammatory Status in Patients with CAD

We further analysed the level of inflammation in the patients by studying a panel of eight cytokines: IL-2, IL-6, IL-8, IL-10, IL-15, MCP-1, TGF-β, and fractalkine (FKN) (Figure 5). This analysis showed that compared with HD, CAD patients had higher concentrations of IL-15 and FKN, but lower levels of IL-8 and MCP-1. These differences were also observed after splitting the patient group into iCAD and ASCAD. Furthermore, iCAD patients, but not ASCAD patients, had reduced levels of TGF-β. Between the iCAD and ASCAD patients, we noticed as well a difference in IL-10 which was elevated in the latter, although it did not reach statistical significance.

### 2.4. Predictive Model

A hypothetical six-colour flow cytometry panel was designed to evaluate potential immunophenotypic biomarkers for distinguishing between HD, iCAD, and ASCAD. The chosen makers for this purpose were CD45, CD3, CD4, CD57, CCR7, and CD45RA, as our results suggested that CD4+ T cells, their memory subsets, and CD57 expression patterns could be critical differentiators of disease status. We employed Random Forest models to assess how effectively these features could distinguish between the disease stages. Four models were calculated including a multi-class model (HD vs. iCAD vs. ASCAD) and three binary models: HD vs. iCAD + ASCAD (Class A), iCAD vs. HD + ASCAD (Class B), and ASCAD vs. HD + iCAD (Class C).

The multi-class model achieved an Accuracy of 0.8636 (95% CI: 0.6509–0.9709), a Balanced Accuracy of 0.8653, and a Kappa of 0.7509, indicating strong discriminative capacity among HD, iCAD, and ASCAD. Each of the three binary models showed high Accuracy on the test subset, particularly the Class B (iCAD vs. HD + ASCAD) and Class C (ASCAD vs. HD + iCAD) models, both yielding an Accuracy of 0.9091 and Specificity of 1.0000. Class A (HD vs. iCAD + ASCAD) attained an Accuracy of 0.8182 and a Balanced Accuracy of 0.8291. Despite all the binary models having similarly high Specificity, Sensitivity varied; Class B correctly identified 60% of the iCAD samples (Sensitivity 0.6), while Class C identified 50% of the ASCAD samples (Sensitivity 0.5000). The AUC values ranged from 0.8125 (Class C) to 0.9556 (Class B), supporting the overall clinical utility of using our hypothetical flow cytometry panel to distinguish between disease subtypes (Figure 6A).

The variable importance score comparison for each model is shown in Figure 6B’s heatmap. Several flow cytometry-derived variables emerged as key contributors. Multi-class model importance scores suggested that CD4+ TCM and CD4+ TEMRA phenotypes are particularly informative for distinguishing HD, iCAD, and ASCAD showing the highest importance scores. Binary models revalidated the importance of the findings from the multiclass model such as CD4+ TCM (Class A) or CD4+ TEMRA cells (Class B) but also highlighted other populations such as total T cell counts (Class A), or the frequency of CD4+ TN and TCM cells expressing the CD57 marker (Class C) (Figure 6B).

## 3. Discussion

In-depth immunophenotyping of CAD patients revealed a combination of shared immune dysregulation across CAD as a whole and subgroup-specific differences between iCAD and ASCAD. While CAD patients exhibited immune activation and senescence-associated changes, further analysis demonstrated that iCAD and ASCAD represent two distinct immune profiles with different underlying mechanisms.

The results presented include both absolute cell counts and percentages, as both metrics provide complementary insights: absolute values capture overall changes in cell numbers, whereas percentages highlight relative shifts within immune subsets. This combined approach offers a more nuanced understanding of immune dysregulation in CAD and ASCAD, particularly given the variability in total leukocyte counts among individuals.

### 3.1. General Immune Dysregulation in CAD

When CAD patients were considered as a whole, several immune alterations were observed compared to HD, reflecting the general immune dysregulation in the disease. However, not all these alterations were statistically significant in both iCAD and ASCAD patients. Regardless of the presence of AS, CAD patients exhibited a significant reduction in the proportion and counts of several innate immune subpopulations, including DCs, primarily due to a decline in pDCs, as well as basophils. A significant reduction in circulating pDCs has been previously reported in CAD patients, highlighting their role in immune regulation and atherosclerosis progression [20]. However, contrarily to Kott et al., our results demonstrated a decline in B cell, T cell (CD4+, CD8+, Tregs), and NK cell (CD56^dim^) counts, suggesting a broader impact on both adaptive and innate immunity. This discrepancy may be due to differences in methodology, as we analysed fresh whole blood rather than PBMCs.

We further observed a general decline in the lymphocyte proportion, that was more pronounced in the ASCAD group in which lymphocyte counts were additionally reduced. Interestingly, NK cell expressing CX3CR1 counts were decreased in patients compared to HD, suggesting that these cells’ decline could reflect their migration to the vasculature [9]. These cells’ proportion was also reduced but not significantly.

The analysis of T cell phenotypes showed an association between CAD and the loss of CD28, CD27, CD25, and CD56 by CD4+ T cells, and a reduction in the TN and TCM phenotypes. A similar decline was observed for CD8+ T cells, affecting CD28, CD27, CD127, CCR7, and CD45RA expression. This supports our observation—consistent with previous studies [14,20] and single-cell transcriptomic data [21]—of an expansion of cytotoxic and exhausted CD57⁺CD28⁻ CD8⁺ T cells, marked by the upregulation of Granzyme B, Perforin, and TCR signalling genes like ZAP70 and CD3E, and associated with chronic activation, impaired proliferative capacity, and vascular injury. Additionally, we identified a loss of costimulatory molecules and IL-2R in TCRγδ cells, along with an increased proportion of the TEMRA phenotype in CAD patients. Taken together, these findings support the notion that CAD is driven by chronic immune activation (continuous TCR engagement), leading to the accumulation of terminally differentiated, pro-inflammatory T cell subsets and progressive immune dysfunction. Additionally, our results indicate significant cytokine alterations in CAD, highlighting the interplay between inflammation, immunity, and atherosclerosis progression. The role of FKN (CX3CL1) in monocyte and lymphocyte adhesion to the endothelium, combined with the CX3CR1+ NK cells decline in CAD patients’ periphery, suggesting an active process of immune cell migration and retention within the vasculature [9]. In line with this, increased IL-15 levels play a distinct role in CAD pathogenesis by driving the expansion and function of pro-inflammatory CD28^null^ T cells, which have been linked to chronic inflammation and plaque instability [22]. These cells exhibit an enhanced cytotoxic phenotype and are resistant to conventional regulatory mechanisms, perpetuating the immune dysfunction and chronic inflammation observed in CAD. Furthermore, enhanced chemokine-driven migration and functional alterations of Tregs has been shown in CAD patients, reflecting a potentially dysfunctional adaptive regulatory response in CAD [20]. In this regard, our data show a decline in Treg counts in iCAD patients that is further aggravated in the presence of AS. Thus, targeting FKN-CX3CR1 or IL-15 pathways may offer therapeutic strategies against immune-mediated vascular damage in CAD.

The concomitant decrease in IL-8 and MCP-1 presents an intriguing paradox, as these cytokines play a critical role in atherosclerosis progression by mediating neutrophil and monocyte recruitment to inflamed tissues. Murine models indicate that KC (IL-8 mouse equivalent) facilitates the initial adhesion of leukocytes to the endothelium, while MCP-1/CCR1 are essential for transendothelial migration and subendothelial infiltration [23,24]. Our data discrepancy with previous studies, such as those by Romuk et al. and Boekholdt et al., may be attributed to the differences in the cohort composition and disease stage [25,26]. Romuk et al. observed elevated IL-8 levels in unstable coronary disease, where acute inflammation drives leukocyte recruitment and plaque destabilisation. In contrast, our cohort comprises patients with advanced but stable CAD, where chronic immune dysregulation, rather than acute inflammatory activation, is more prominent. Additionally, the observed decline in CX3CR1+ NK cell counts and the elevated NLR in ASCAD indicate the preferential recruitment of immune cells to plaques, potentially depleting systemic MCP-1. Moreover, the accumulation of pro-inflammatory CD57+ CD28^null^ T cells in CAD promotes chronic inflammation, likely sustaining local MCP-1 and IL-8 production within the vasculature while reducing their circulating levels. Chronic immune activation may also induce compensatory feedback suppression, limiting further monocyte and neutrophil recruitment into the circulation. Collectively, these findings suggest a shift from systemic inflammation towards a more localised vascular and/or valvular inflammatory response, driven by immune cell sequestration, senescence-driven dysregulation, and compensatory anti-inflammatory mechanisms.

Considering these findings, IL-15 and FKN emerge as potential therapeutic targets in CAD. IL-15 plays a key role in T cell proliferation and cytotoxic activity, and several IL-15-based therapies, including superagonists (e.g., ALT-803, NKTR-255) and receptor antagonists, are under development, mainly for oncological and autoimmune indications [27]. Likewise, FKN, which mediates immune cell adhesion and migration, is being targeted in inflammatory diseases through agents such as the CX3CR1 antagonist, KAND567, and the monoclonal antibody, E6011 [28]. Although none of these therapies have yet been evaluated specifically in CAD, our results support the rationale for exploring the therapeutic potential of IL-15 or FKN modulation in future cardiovascular trials.

These alterations observed in both groups of patients align with the concept that atherosclerosis and AS share common inflammatory and immunological pathways, with endothelial injury and chronic inflammation as key triggers of both atherosclerotic plaque formation and valvular calcification [29]. The accumulation of immune cells and cytokines within vascular and valvular tissues contributes to disease progression, reinforcing the notion of CAD as a systemic inflammatory disorder rather than a strictly vascular condition. Understanding these mechanisms is critical for developing immunotherapies aimed at mitigating chronic inflammation and vascular damage in CAD patients. The overlap in immune dysfunction seen in iCAD and ASCAD patients suggests that targeting chronic inflammation may be a shared therapeutic strategy. Nevertheless, our results demonstrate the existence of key differences between iCAD and ASCAD, defining them as distinct clinical entities.

### 3.2. Distinct Immune Features of iCAD: A Predominantly Activated Immune State

Isolated CAD (iCAD) patients exhibited a more pro-inflammatory and immune-activated profile compared to ASCAD. Particularly, in iCAD, but not ASCAD, there was a higher proportion of inflammatory CD14+CD16+ monocytes, suggesting that iCAD is associated with heightened monocyte activation and inflammation, potentially driving disease progression through endothelial activation and vascular remodelling. Additionally, iCAD patients had higher proportions of CD56^bright^ cells and higher activation of both the CD56^bright^ and CD56^dim^ subsets (HLA-DR expression). At the T cell level, iCAD patients exhibited higher proportions of activated (HLA-DR+) CD4+ and CD8+ T cells, further supporting the idea that iCAD is characterised by heightened immune activation rather than pure immune ageing.

Furthermore, iCAD patients had higher frequencies of Tregs, but a reduction in TGF-β, which is a key regulator of Tregs. Its decline in iCAD may reflect a loss of immune homeostasis, leading to chronic inflammation. Additionally, TGF-β plays a crucial role in endothelial integrity and tissue remodelling, and its downregulation may hinder vascular repair, accelerating plaque destabilisation. In contrast, ASCAD patients TGF-β levels are less reduced, and Treg proportion maintained. The accumulation of immunosenescent T and NK cells may be compensated for by secreting anti-inflammatory factors. Indeed, although nonsignificant, ASCAD patients showed elevated levels of IL-10. This distinction reinforces the notion that iCAD is driven by excessive immune activation, while ASCAD represents a more advanced stage of immunosenescence and immune exhaustion.

### 3.3. Distinct Immune Features of ASCAD: A Predominantly Immunosenescent and Cytotoxic Profile

In contrast to iCAD, ASCAD patients exhibited more pronounced immune ageing, cytotoxicity, and a loss of regulatory control, suggesting that ASCAD is not merely an advanced form of CAD but a distinct immunological entity with unique immune drivers. One of the most prominent immune shifts in ASCAD was a greater reduction in total lymphocyte counts, contributing to a significantly higher NLR and lower LMR compared to iCAD. This suggests that ASCAD is associated with a greater systemic inflammatory burden. A higher NLR and platelet-to-lymphocyte ratio (PLR) have been associated with higher inflammatory activity and worse CAD prognosis [30], supporting our observations and reinforcing the connection between chronic inflammation, immune imbalance, and disease severity.

At the T cell level, ASCAD patients displayed a stronger immunosenescence signature, highly enriched for CD57+ and CD28^null^ T cells. The accumulation of these cells, particularly within the CD4+ T cell subset, has been associated with disease severity [31]. Previous results from our group have also demonstrated the presence of these proatherogenic cells in valve infiltrates [32]. This was in sharp contrast to iCAD, where CD4+ T cells were more activated (HLA-DR+) rather than purely senescent and cytotoxic. Additionally, ASCAD patients exhibited a higher proportion of cytotoxic NK cells, with a significant increase in CD57+ NK cells, reinforcing a more aggressive and tissue-damaging immune profile. This was further supported by a higher mean fluorescence intensity (MFI) of CD57 in NK and T cells, suggesting a stronger per-cell expression of senescence and cytotoxicity markers. Furthermore, unlike iCAD patients, who exhibited an increased Treg percentage, ASCAD patients had similar or even lower (trend) Treg frequencies than HD, indicating a failure of immune regulation. This loss of regulatory control could contribute to the higher levels of immune activation and tissue damage in these patients. In line with these findings, higher systemic immune-inflammation response index (SIIRI) levels have been associated with an increased incidence of major adverse cardiovascular events, reinforcing the role of chronic inflammation in driving immune exhaustion and dysfunction [30]. Consistently, our results demonstrate that ASCAD is characterised by a predominant immunosenescent and cytotoxic profile, with a greater accumulation of highly differentiated CD57+ and CD28^null^ T cells, as well as cytotoxic NK cells, highlighting the stronger role of immunosenescence in CAD when associated with aortic stenosis.

### 3.4. Immunosenescence as a Key Driver of ASCAD Pathophysiology

The immune landscape observed in ASCAD highlights immunosenescence as a key mechanism driving cardiovascular disease progression. While immunosenescence features are also present in iCAD patients, they appear to be more pronounced in ASCAD, where severer lymphopenia, loss of regulatory T cells, and the accumulation of cytotoxic T and NK cells (CD28^null^, CD57+) define a more severe immune dysfunction. A major contributor to T cell senescence is chronic antigenic stimulation, and growing evidence suggests that latent viral infections, particularly cytomegalovirus (CMV), play a significant role in this process. More than two decades ago, Jonasson et al. showed that CAD patients displayed an expansion of CD8+ CD57+CD28^null^ T cells, and that both CMV seropositivity and CAD were independently associated with this expansion, suggesting that both chronic viral antigen exposure and CAD-related inflammation act synergistically to drive immunosenescence [33].

The observed elevation of IL-10 in ASCAD, where CD28^null^ T cells are most expanded, presents a potential compensatory mechanism. Unlike previous studies in chronic heart failure, where IL-10 was associated with a reduction in CD4+CD28^null^ T cell frequencies, in ASCAD, its increase appears insufficient to counteract the inflammatory burden imposed by accumulated senescent T cells. Given that IL-10 inhibits T cell activation through costimulatory blockade (LFA-3/CD2 and HLA class II downregulation) [34], its upregulation in ASCAD might reflect an attempt to suppress excessive inflammation rather than an effective immunosuppressive response. This further supports the idea that ASCAD represents a phase of immune exhaustion, rather than ongoing immune activation.

In a previous study, we found that CMV infection perpetuates the permanent loss of CD28 in CD4+ T cells, independently of age, and is the main driver of pro-atherogenic CD4+ CD28^null^ T cell expansion [35]. Moreover, our recent cytometry-based analysis of calcified aortic valves revealed the presence of these proatherogenic CD4+ CD28^null^ T cells in the valvular infiltrate of AS patients, a finding of particular relevance for ASCAD, where CD4+ CD28^null^ T cells accumulate [32]. These cells foster vascular damage, endothelial dysfunction, and chronic inflammation [36], suggesting that persistent CMV infection may compound the immune dysregulation seen in CAD-associated aortic stenosis. As the seroprevalence of CMV was similar between iCAD and ASCAD patients (Appendix A), the observed differences seem to be more likely related to the immune response to the virus rather than to the infection per se. Thus, systemic immune alterations, aggravated by chronic viral antigen exposure, could accelerate immune ageing in ASCAD. Comprehensive analyses of peripheral immune profiles and valvular infiltrates are therefore vital to understand the overlapping and distinct immunopathological mechanisms of CAD and AS, informing more targeted immunotherapies.

Despite these insights, it remains unclear whether the immunological changes in ASCAD directly cause aortic stenosis or if they primarily characterise the combined presence of AS and CAD. To resolve this, future studies should examine AS in the absence of CAD, discerning which immune pathways are unique to valvular disease versus those linked to atherosclerosis-related inflammation. Clarifying these independent contributions is crucial for designing immunomodulatory therapies for AS, particularly in patients without coexisting CAD.

### 3.5. Potential Cytometry Panel for Disease Distinction

Our findings demonstrate that a simple six-colour flow cytometry panel targeting CD45, CD3, CD4, CD57, CCR7, and CD45RA could provide robust discrimination among healthy donors and patients at different stages of coronary artery disease (iCAD vs. ASCAD). The predictive models achieved high accuracy and specificity, particularly in the binary comparisons of iCAD or ASCAD against the other groups. The multi-class model further underscored the potential clinical value of specific T cell phenotypes, with several variables such as CD4+ TCM cells (frequency and absolute numbers) and CD4+ TEMRA cells emerging as the strongest predictors.

From a clinical perspective, these observations support the notion that CD4+ T cell memory subsets and CD57 expression patterns could be critical for identifying individuals at risk or monitoring disease progression. While sensitivity varied across the binary models—especially in distinguishing ASCAD—this suggests that integrating additional variables or refining the panel could further enhance the detection rates. Ultimately, these results provide evidence that immunophenotypic profiling provides essential insights into the physiopathology of this disease that will set the grounds for the development of novel and more personalised interventions in coronary artery disease management.

**Limitations:** A key limitation of this study is its cross-sectional design, which restricts our ability to assess the temporal changes in immune cell profiles and their relationship with disease progression. This also limits our capacity to determine whether the observed differences between iCAD and ASCAD reflect the sequential stages of disease or rather distinct immunological susceptibilities inherent to each patient group. Longitudinal studies with serial immune profiling will be essential not only to elucidate the dynamics of immune alterations over time but also to clarify whether immune signatures evolve during disease progression or are fixed traits. Additionally, the relatively small and homogeneous sample size may limit the generalisability of our findings, and caution is warranted when extrapolating our results to broader populations. Future studies should aim to validate the identified immune phenotypes in larger, multi-centre cohorts, particularly in earlier or moderate stages of disease. Furthermore, although we excluded patients treated with calcium channel blockers due to their well-described immunosuppressive effects [37], other antihypertensive drugs were not considered as exclusion criteria. While there is evidence that ACE inhibitors, angiotensin receptor blockers, beta-blockers, and thiazide diuretics can modulate immune responses, these effects appear modest and heterogeneous, and are mostly supported by preclinical studies [38,39]. We acknowledge that not accounting for these treatments represents a limitation of our study, and future work should aim to stratify or control for antihypertensive therapy to better assess its potential influence on immune phenotyping in CAD.

## 4. Materials and Methods

### 4.1. Study Design and Participants

A prospective study was designed for patient recruitment. All patients were selected from Reina Sofía University Hospital’s targeted population. Participant selection was performed according to the following criteria, defining two groups of patients: those with severe (plaque-induced stenosis between 50% and 99%) isolated coronary artery disease (iCAD) and those with associated calcific aortic stenosis (tricuspid valve) and an indication for surgical replacement (ASCAD). Cardiac catheterisation and echocardiography were performed in every participant by the Cardiology Unit. Current ESC/EACTS guidelines were used to define the severity. The control group—with equivalent epidemiological factors—comprised healthy volunteers without any cardiovascular or inflammatory-based disease (HDs). Before inclusion, all individuals underwent a clinical and epidemiological characterisation. Interviews to harvest information regarding comorbidities and a thorough examination of individuals’ medical history were performed. This included gender, date of birth (age), weight, height, BMI, diabetes, hypertension, obesity, dyslipidaemia, chronic renal disease, Chronic Obstructive Pulmonary Disease, stroke, smoking, alcohol, and drug intake. Equal exclusion criteria were applied to all groups regarding comorbidities: chronic infectious diseases (HCV, HBV, HIV), acute infection when a blood sample was taken, Cancer, immunological diseases (rheumatoid arthritis, ankylosing spondylitis, ulcerative colitis, Crohn’s disease, celiac disease, Systemic Lupus Erythematosus, multiple sclerosis, psoriasis, etc.), haemodialysis, immunosuppressants drugs, and dihydropyridine calcium-channel blockers.

A total of 72 participants were included in the study (Appendix A). Peripheral blood samples were collected by the Cardiovascular Surgery Unit.

### 4.2. Ethics Approval

All participants were informed, and signed informed consent was obtained to participate in the study. The study was approved by the Ethics Committee of Hospital Universitario Reina Sofia of Cordoba (Spain) (Immunopathology of aortic stenosis, Committee reference number 4047, 19 December 2019).

### 4.3. Sample Collection, Processing, and Data Acquisition

Peripheral blood was collected from each participant in EDTA tubes through venipuncture. For the immunophenotyping, whole fresh blood (100 μL) was stained with monoclonal, fluorescent-labelled antibodies (Appendix A). Cells were incubated for 20 min at room temperature in the dark and lysed with FACS Lysing Buffer (BD Becton Dickinson), according to the manufacturer’s instructions. Samples were acquired within 1–4 h in a BD LSR Fortessa SORP cytometer. Spectral overlap compensation between all channels was performed automatically by using the BD FACSDiva software v9.0 (BD Biosciences, Milpitas, CA, USA), using single-colour controls. For the standardisation of instrument settings longitudinally, an 8-peak Rainbow Compensation Particles Set (BD) was used before every experiment, and the photomultiplier tube voltages were adjusted if needed.

### 4.4. Processing Data, Modelling Process, and Statistical Analysis

Flow cytometry data were analysed using FlowJo v10.10.0 (TreeStar, Portland, OR, USA). The FlowJo Boolean gating tool was used to create co-expression profiles. The gating strategy is shown in Appendix A.

Immune cell data are presented both as absolute counts (cells/μL) and as percentages relative to the parent populations. This dual approach was adopted to reflect both the quantitative and compositional changes in circulating immune cells. Absolute counts are particularly informative for assessing global immune cell depletion or expansion (e.g., lymphopenia, neutrophilia), while percentages allow for the detection of shifts in immune subpopulation balance, which may be masked by changes in the total cell numbers. This strategy enables a more comprehensive interpretation of immune alterations across disease groups.

Prediction models were conducted in R to evaluate immune and clinical biomarkers across disease stages. Four Random Forest classification models were constructed in R version 4.2.3 using the randomForest package. The dataset was randomly split into a training subset (70% of the samples) and a test subset (30% of the samples). Each model was built with 20.000 trees, and the number of variables tried at each split (mtry) was determined via internal tuning. One model was multi-class (HD vs. iCAD vs. ASCAD), and three were binary: HD vs. (iCAD + ASCAD) (Class A), iCAD vs. (HD + ASCAD) (Class B), and ASCAD vs. (HD + iCAD) (Class C). The models’ performance was evaluated on the test set using Accuracy (with 95% confidence intervals), Cohen’s Kappa, Balanced Accuracy, Sensitivity, Specificity, and Area Under the ROC Curve (AUC). Given the limited sample size, we relied on the out-of-bag (OOB) error estimate provided by the Random Forest algorithm as an internal form of cross-validation. This method evaluates the model performance on data not used during tree construction and is well suited for small datasets, helping reduce the risk of overfitting without further partitioning the data. Variable importance scores were obtained with the ‘importance’ function of the Random Forest library algorithm and visualised in a heatmap for comparative analysis of each variable’s contribution across the four models. This function estimates the contribution of each variable to the model’s performance through a mean decrease in accuracy (MDA) calculation. The MDA is calculated by randomly permuting each variable and measuring the corresponding drop in classification accuracy for the OOB data.

For statistical analysis, normal distributions were calculated using the Shapiro–Wilk test, and the significance of data was determined using the Kruskal–Wallis test, followed by the Mann–Whitney comparison test using R software (version 4.3.1; https://www.r-project.org, accessed on 15 October 2024). Analysis of T and NK cell phenotype profiles was performed using SPICE software (version 6.1; https://niaid.github.io/spice/, accessed on 18 November 2024) developed by Mario Roederer (ImmunoTechnology Section at the Vaccine Research Centre, National Institutes of Health). To compare the pie charts, we used the SPICE permutation test. This analysis determines the likelihood of observing the difference between the 2 pie charts by chance, given the samples that comprise them, through 10,000 permutations. Individual pie slices were compared using the Wilcoxon rank test. GraphPad Prism software (version 8.0; GraphPad Software) was used for graph building.

## 5. Conclusions

This study highlights distinct immune signatures in CAD patients, demonstrating that while iCAD and ASCAD share some common immune features, they are two separate immunological entities (Figure 7). The data suggest that iCAD is predominantly driven by heightened immune activation, while ASCAD is characterised by an advanced immunosenescent and cytotoxic immune profile, probably enhanced by chronic viral infection. These findings challenge the traditional view of ASCAD as merely a more severe form of CAD and suggest that it represents a distinct immune-mediated disease. This distinction highlights the importance of considering immune ageing as a therapeutic target, particularly in ASCAD patients, to mitigate disease progression and improve the clinical outcomes.

## Figures and Tables

**Figure 1 ijms-26-05248-f001:**
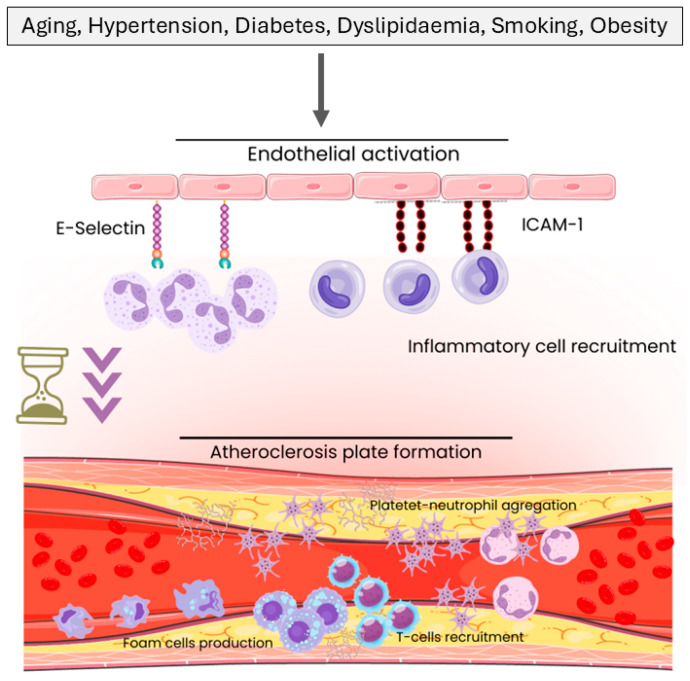
Schematic representation of the key risk factors and cellular mechanisms involved in atherosclerotic plaque formation. Clinical risk factors such as ageing, hypertension, diabetes, dyslipidaemia, smoking, and obesity contribute to endothelial activation, inflammatory cell recruitment, platelet–neutrophil aggregation, foam cell production, and T cell recruitment, ultimately leading to the development of atherosclerotic plaques.

**Figure 2 ijms-26-05248-f002:**
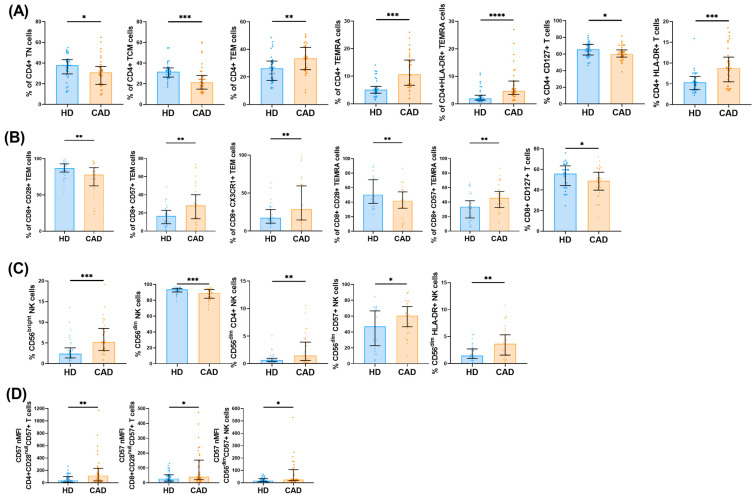
T and NK cell phenotype alterations in CAD patients. Bar graphs showing the frequencies of (**A**) CD4+, (**B**) CD8+ T cells, (**C**) NK cells, and (**D**) bar graphs showing CD57 nMFI in T and NK cytotoxic cells. The top of the bars indicates the median, and whiskers show the IQR, ranging from the 25th to the 75th percentile. Statistical significance: * *p* < 0.05, ** *p* < 0.01, *** *p* < 0.001, and **** *p* < 0.0001.

**Figure 3 ijms-26-05248-f003:**
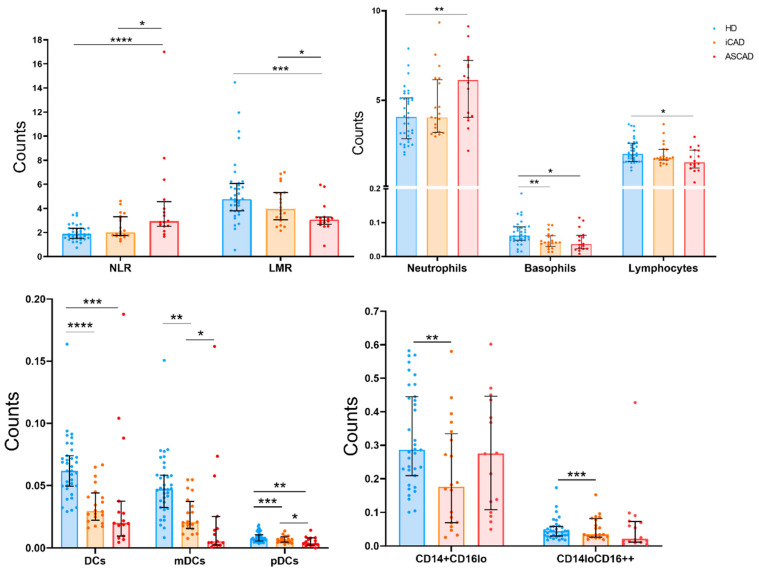
Immune alterations in iCAD and ASCAD patients. Bar graphs showing the absolute numbers of immune populations altered differently depending on AS. The top of the bars indicates the median, and whiskers show the IQR, ranging from the 25th to the 75th percentile. Statistical significance: * *p* < 0.05, ** *p* < 0.01, *** *p*< 0.001, and **** *p* < 0.0001.

**Figure 4 ijms-26-05248-f004:**
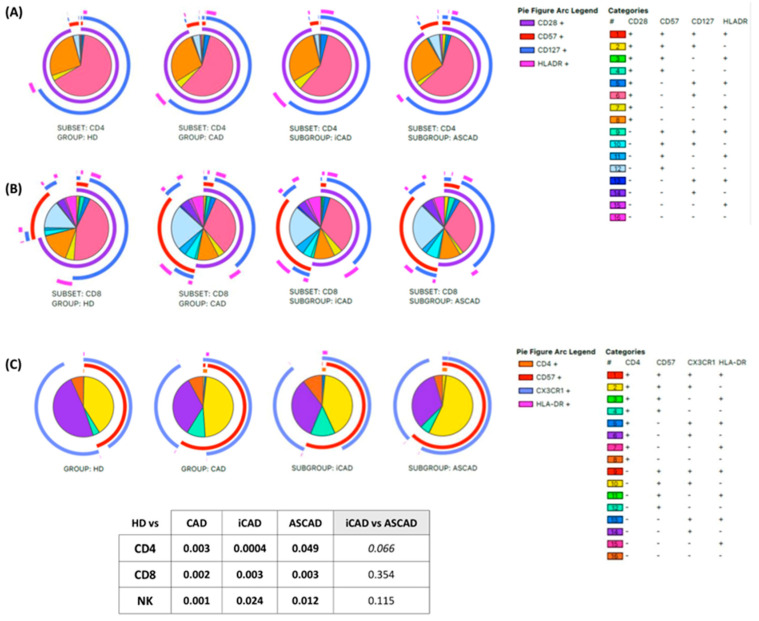
T and NK cell phenotypic profiles. Pie charts display the phenotypic profiles of (**A**) CD4⁺ T cells, (**B**) CD8⁺ T cells, and (**C**) NK cells in HD and patient groups. The pie arc and slice legends are shown on the right. A plus sign (+) indicates marker expression, while a minus sign (−) indicates the absence of expression. The table at the bottom presents statistical significance values for comparisons between groups (SPICE permutation test). Italics indicate tendency.

**Figure 5 ijms-26-05248-f005:**
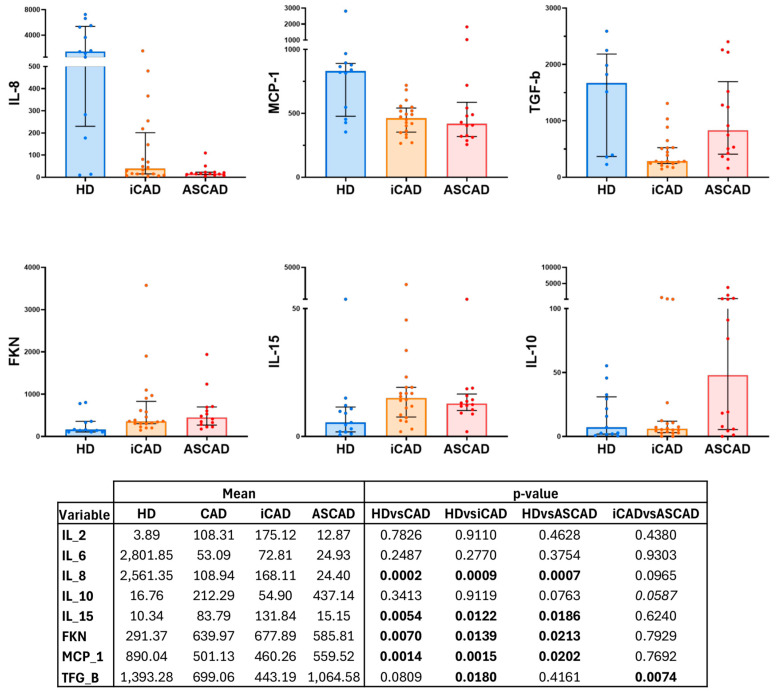
Cytokine analysis in CAD patients. The graphs show alterations in cytokine levels in iCAD and ASCAD patients. The top of the bars indicates the median, and whiskers show the IQR, ranging from the 25th to the 75th percentile. Statistical significance for each cytokine measured is indicated in the table below. Significant values are shown in bold. Italics indicate tendency.

**Figure 6 ijms-26-05248-f006:**
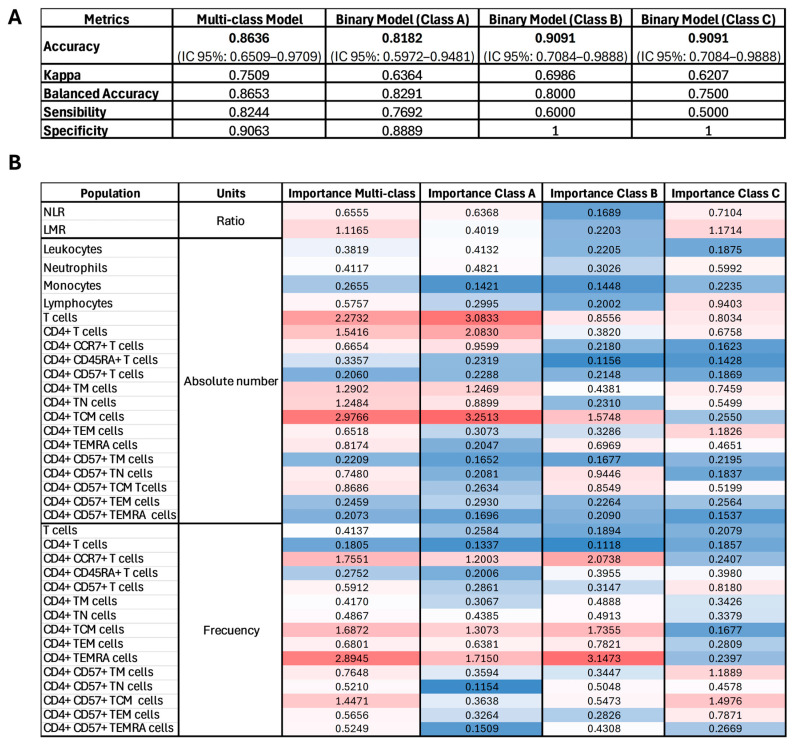
Random Forest model performance and variable importance. (**A**) Model performance metrics table for each model, including Accuracy, Cohen’s Kappa, Balanced Accuracy, Sensitivity, Specificity, and Area Under the ROC Curve (AUC). (**B**) Heatmap of variable importance of multi-class, and Class A, Class B, and Class C binary models. Warmer (red) hues represent higher importance scores, indicating variables that most effectively discriminate among the classes; cooler (blue) hues denote lower importance.

**Figure 7 ijms-26-05248-f007:**
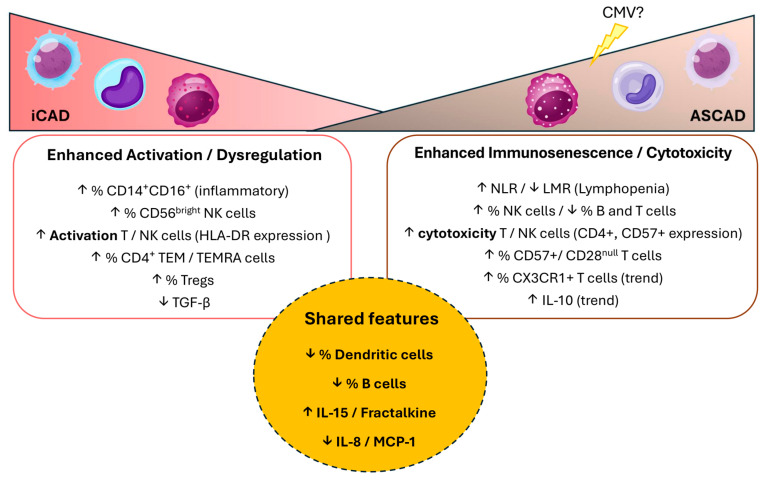
Summary of immune alterations in iCAD and ASCAD. Schematic representation of the key immunological differences and shared features between iCAD and ASCAD patients. ↑ denotes an increase, ↓ denotes a decrease. Bolded terms indicate particular emphasis noted in the data. The dashed outline around the central yellow area marks features shared between both conditions. The question mark next to “CMV?” suggests a potential, but not confirmed, involvement of cytomegalovirus in the observed immune alterations.

**Table 1 ijms-26-05248-t001:** Absolute counts of major circulating immune cell populations across study groups.

	Absolute Numbers							
	Mean				*p*-Value			
	HD	CAD	iCAD	ASCAD	HD vs. CAD	HD vs. iCAD	HD vs. ASCAD	iCAD vs. ASCAD
LEUKOCYTES	7.1061	8.0683	7.8380	8.3563	*0.0561*	0.2141	0.0703	0.4875
NEUTROPHILS	4.1519	5.1683	4.6815	5.7769	**0.0250**	0.3514	**0.0069**	0.0620
BASOPHILS	0.0696	0.0468	0.0464	0.0474	**0.0016**	**0.0043**	**0.0251**	0.9164
MONOCITES	0.4403	0.5139	0.5050	0.5250	0.1424	0.1928	0.1493	0.5772
LYMPHOCYTES	2.1608	1.8561	1.9975	1.6794	**0.0474**	0.3129	**0.0209**	0.1387
NLR	1.9925	3.2962	2.4635	4.3370	**0.0006**	0.0957	**0.0000**	**0.0215**
LMR	5.3645	3.8324	4.2943	3.2551	**0.0026**	0.1401	**0.0003**	**0.0457**
CD4:CD8 Ratio	2.9422	2.4564	2.3919	2.5369	0.3668	0.4399	0.4979	0.7410
DCs	0.0643	0.0370	0.0348	0.0400	**0.0000**	**0.0000**	**0.0005**	0.1794
mDCs	0.0497	0.0259	0.0261	0.0256	**0.0000**	**0.0040**	0.6263	**0.0277**
pDCs	0.0085	0.0060	0.0067	0.0050	**0.0067**	**0.0001**	**0.0003**	**0.0253**
dn DCs	0.0047	0.0045	0.0019	0.0081	0.0977	0.0982	**0.0040**	0.1438
CD14+CD16lo	0.3231	0.2770	0.2086	0.3626	**0.0295**	**0.0100**	0.2069	0.1317
CD14+CD16+	0.0495	0.0566	0.0525	0.0618	0.5406	0.8828	0.3922	0.1491
CD14loCD16++	0.0088	0.0054	0.0041	0.0071	**0.0021**	**0.0010**	0.1071	0.6258
B cells	0.3657	0.1578	0.1613	0.1534	**0.0000**	**0.0000**	**0.0000**	0.7413
NK cells	0.4136	0.2610	0.2173	0.3157	**0.0001**	**0.0000**	**0.0201**	**0.0389**
T cells	1.9152	1.2096	1.2688	1.1355	**0.0000**	**0.0000**	**0.0001**	0.4565
CD4	1.2864	0.7546	0.8371	0.6515	**0.0000**	**0.0000**	**0.0000**	0.0957
CD8	0.5472	0.3981	0.3755	0.4263	**0.0004**	**0.0057**	**0.0026**	0.0827
CD4hiCD8lo	0.0349	0.0136	0.0122	0.0152	0.1652	**0.0370**	0.9765	0.0949
DP	0.0131	0.0068	0.0070	0.0065	**0.0018**	**0.0215**	**0.0048**	0.3049
DN	0.0682	0.0501	0.0492	0.0512	**0.0188**	0.1065	**0.0246**	0.7176
DN TCRγδ	0.0567	0.0397	0.0403	0.0390	**0.0361**	0.1985	**0.0289**	0.4984
DN TCRab	0.0116	0.0104	0.0089	0.0122	0.0902	0.0890	0.3202	0.9624
T reg	0.0829	0.0540	0.0659	0.0390	**0.0001**	**0.0256**	**0.0000**	**0.0039**

Mean absolute cell numbers (cells/μL) and corresponding *p*-values comparing healthy donors (HDs), patients with isolated coronary artery disease (iCAD), and those with associated aortic stenosis (ASCAD), *p*-values ≤ 0.05 are highlighted in bold; *p*-values close to significance (*p* = 0.05) are highlighted in italics.

**Table 2 ijms-26-05248-t002:** Relative proportions (%) of major circulating immune cell populations across study groups.

	Percentages							
	Mean				*p*-Value			
	HD	CAD	iCAD	ASCAD	HD vs. CAD	HD vs. iCAD	HD vs. ASCAD	iCAD vs. ASCAD
NEUTROPHILS	57.6645	63.1828	59.2507	68.0978	**0.0208**	0.5958	**0.0008**	**0.0188**
BASOPHILS	0.8164	0.6069	0.6027	0.6122	**0.0121**	0.0646	**0.0238**	0.4402
MONOCITES	6.3300	6.5400	6.4982	6.5922	0.6704	0.8337	0.6224	0.8789
LYMPHOCYTES	30.9218	23.8535	26.2981	20.7977	**0.0001**	**0.0310**	**0.0001**	0.0949
DCs	0.7534	0.4980	0.4545	0.5561	**0.0000**	**0.0000**	**0.0062**	0.0743
mDCs	0.5821	0.3412	0.3388	0.3443	**0.0000**	**0.0104**	0.7348	0.1222
pDCs	0.0997	0.0793	0.0892	0.0662	0.0643	**0.0003**	**0.0008**	**0.0143**
dn DCs	0.0551	0.0691	0.0245	0.1285	0.1268	0.3629	**0.0262**	0.1470
CD14+CD16lo	79.8951	76.7538	72.6970	81.8248	0.5256	*0.0526*	0.4035	**0.0010**
CD14+CD16+	13.0839	18.9086	24.3082	12.1591	0.0736	**0.0004**	0.2717	**0.0179**
CD14loCD16++	2.2437	1.6752	1.7062	1.6364	**0.0242**	*0.0596*	**0.0280**	0.2114
B cells	4.4097	2.0638	2.2182	1.8708	**0.0000**	**0.0000**	**0.0000**	0.5819
NK cells	12.3130	16.0074	13.6464	18.9587	0.0774	0.4093	**0.0312**	0.1783
T cells	70.2394	68.1008	70.4790	65.1281	0.4838	0.8952	0.1828	0.2358
CD4	67.3019	63.7722	65.3295	61.8256	0.3974	0.4891	0.4493	0.9124
CD8	28.2339	31.3919	29.8570	33.3106	0.4174	0.5126	0.6062	0.8381
CD4hiCD8lo	2.7153	2.0528	1.6825	2.5156	0.5468	0.3382	**0.0292**	**0.0071**
DP	0.6736	0.5498	4.2526	0.5349	0.2408	0.9932	0.2330	0.7652
DN	3.7772	4.2866	80.3963	4.3290	0.7076	0.6903	0.4979	0.7176
DN TCRγδ	78.8747	79.1023	19.6037	77.4848	0.4165	0.6903	0.3399	0.4984
DN TCRab	21.1253	20.8977	0.5618	22.5152	0.4165	0.4603	0.3399	0.4984
T reg	6.5278	7.2093	8.1446	6.0401	0.1584	**0.0057**	0.4828	**0.0149**

Mean percentage and corresponding *p*-values comparing healthy donors (HDs), patients with isolated coronary artery disease (iCAD), and those with associated aortic stenosis (ASCAD), *p*-values ≤ 0.05 are highlighted in bold; *p*-values close to significance (*p* = 0.05) are highlighted in italics.

## Data Availability

The data that support the findings of this study are available from the corresponding author, A.P., upon reasonable request.

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
