# Peer review of "Divergent Immune Pathways in Coronary Artery Disease and Aortic Stenosis: The Role of Chronic Inflammation and Senescence"

_ijms, 2025, doi:10.3390/ijms26115248_

Round 1

Reviewer 1 Report

Comments and Suggestions for Authors

The manuscript is very interesting but I would like to issue several suggestions.

In the Introduction there are two consecutive paragraphs starting with Furthermore which is repetitive.

In the main text, the first reference is 23, then 30. They should be in order from 1.

A figure would be useful in the introduction to better illustrate the pathophysiology of atherosclerosis.

The references should be written in the MDPI format.

Table 1 is far too long, maybe it should be split in several tables.

Author Response

Comment 1: In the Introduction there are two consecutive paragraphs starting with Furthermore which is repetitive.

Response 1: We thank the reviewer for pointing this out. We have revised the text in the Introduction to remove the repetition. The second occurrence of "Furthermore" has been replaced with alternative phrasing for improved readability.

Comment 2: In the main text, the first reference is 23, then 30. They should be in order from 1.

Response 2: We appreciate this observation and apologise for the mistake. All references have now been reordered to appear sequentially in the text, starting from reference 1.

Comment 3: A figure would be useful in the introduction to better illustrate the pathophysiology of atherosclerosis.

Response 3: As suggested, we have included a new schematic figure in the Introduction (new Figure 1) to illustrate the key steps in the pathophysiology of atherosclerosis. This figure aims to enhance the reader’s understanding of the disease mechanisms discussed. We have renamed the rest of the figures accordingly.

Comment 4: The references should be written in the MDPI format.

Response 4: We have reformatted all references according to the MDPI citation style, as requested.

Comment 5: Table 1 is far too long, maybe it should be split in several tables.

Response 5: We thank the reviewer for this suggestion. Table 1 has now been split into Tables 1 and 2 for absolute numbers and percentages of main subsets respectively, and supplementary Tables S4-7 for NK and T cells phenotype (counts and percentages) to improve clarity and readability. We hope this solution will properly address this issue.

Reviewer 2 Report

Comments and Suggestions for Authors

The work Divergent Immune Pathways in Coronary Artery Disease and Aortic Stenosis: The Role of Chronic Inflammation and Senescence by José Joaquín Domínguez-del-Castillo et al. is a valuable contribution that begins to unravel the immune profiles of CAD and highlights key differences between CAD and ASCAD. Notably, the authors propose a predictive model to evaluate potential immunophenotypic patterns. Despite the relevance of these findings, several revisions would enhance the article's readability and clarity.

The main limitation lies in the presentation of the principal table. It is difficult to follow and extract the relevant data. The authors should consider restructuring the table for improved clarity and coherence.

Furthermore, the rationale behind using absolute cell counts versus percentages remains unclear. The article switches between absolute counts and percentages during the analysis. These two approaches can yield different insights, especially when total cell numbers vary across individuals or groups (the statistical comparisons are different between these categorizations). A discussion justifying the choice—or ideally, a consistent analytical approach—would improve both the methodological rigor and the reader's ability to interpret the findings accurately.

There are also some concerns regarding the patient cohort. Antihypertensive medications are known to have immunomodulatory effects—could these treatments influence the immune phenotype observed in CAD patients? Since most CAD patients are hypertensive, it is important to discuss whether this could have impacted the findings.

Furthermore, as iCAD and ASCAD may represent different stages of the same disease, the timing of diagnosis, follow-up, or treatment could potentially influence the interpretation of results. Although all patients were age-matched, they presented with different disease stages. Whether the observed immune phenotypes reflect disease progression or an intrinsic susceptibility in certain individuals remains unclear. A more in-depth discussion of these limitations would strengthen the article.

It was also very difficult to follow the structure and interpretation of the predictive model. The manuscript would benefit from a clearer explanation of how variable importance was assessed and how these results should be interpreted in the context of disease classification. Clarifying whether these values reflect contribution to model accuracy, effect size, or another metric would significantly aid comprehension.

Additionally, a final graphical summary would help readers track the immune phenotypes associated with CAD, iCAD, and ASCAD.

Author Response

Comment 1: The main limitation lies in the presentation of the principal table. It is difficult to follow and extract the relevant data. The authors should consider restructuring the table for improved clarity and coherence.

Response 1: We thank the reviewer for its valuable input. As mentioned in our response to Reviewer 1, Table 1 has been restructured and divided into multiple smaller tables to enhance clarity and to facilitate interpretation.

Comment 2: Furthermore, the rationale behind using absolute cell counts versus percentages remains unclear. The article switches between absolute counts and percentages during the analysis. These two approaches can yield different insights, especially when total cell numbers vary across individuals or groups (the statistical comparisons are different between these categorizations). A discussion justifying the choice—or ideally, a consistent analytical approach—would improve both the methodological rigor and the reader's ability to interpret the findings accurately.

Response 2: We thank the reviewer for this valuable observation regarding the rationale behind the use of absolute cell counts versus percentages. In response, we have revised the manuscript to address this issue more explicitly. Clarifying paragraphs have been added to both the Materials and Methods (Lines 728–735) and Discussion (Lines 452–456) sections to explain our dual approach.

Absolute counts provide insight into overall immune cell expansions or depletions, while percentages offer a relative view of shifts within immune subpopulations—particularly important when total leukocyte counts vary across individuals.

Furthermore, the Results section and Table 1 have been restructured to improve clarity and coherence. Where applicable, results are now grouped or described in a way that distinguishes between findings based on absolute values and those based on proportions. This restructuring enhances the reader’s ability to interpret the data accurately and understand the implications of each analytical perspective. Table 1 has been split into Table 1 (counts) and 2 (%), and supplementary tables S4-7 (NK and T cells phenotypes).

Comment 3: There are also some concerns regarding the patient cohort. Antihypertensive medications are known to have immunomodulatory effects—could these treatments influence the immune phenotype observed in CAD patients? Since most CAD patients are hypertensive, it is important to discuss whether this could have impacted the findings.

Response 3: We thank the reviewer for raising this important point. Indeed, the immunomodulatory effects of antihypertensive therapies are increasingly recognized. In our cohort, a high proportion of patients with CAD were hypertensive, consistent with clinical expectations. While we did not stratify patients based on the specific antihypertensive medications received, we deliberately excluded calcium channel blockers (CCBs) from our study due to their known suppressive effects on lymphocyte proliferation and cytokine production (1-3), which could have confounded immune phenotyping results.

Other classes of antihypertensives, such as ACE inhibitors (ACEIs), angiotensin receptor blockers (ARBs), beta-blockers, and thiazide diuretics, may exert modulatory effects on immune cells; however, current literature suggests that these effects are either modest, inconsistent, or primarily derived from animal models or acute pathophysiological settings. To date, there is insufficient evidence indicating that these treatments significantly alter the circulating immune cell subsets we analysed, particularly in stable CAD patients.

Nonetheless, we acknowledge that this is a limitation of our study and agree that future research should aim to stratify patients based on antihypertensive regimens to better understand their impact on immune phenotypes. We have included this consideration in the revised version of the manuscript (limitations) at the end of the discussion section.

1) 10.3390/pharmaceutics14071478

2) 10.1038/ajh.2007.13

3) 10.1016/j.imlet.2016.02.003

Comment 4: Furthermore, as iCAD and ASCAD may represent different stages of the same disease, the timing of diagnosis, follow-up, or treatment could potentially influence the interpretation of results. Although all patients were age-matched, they presented with different disease stages. Whether the observed immune phenotypes reflect disease progression or an intrinsic susceptibility in certain individuals remains unclear. A more in-depth discussion of these limitations would strengthen the article.

Response 4: We appreciate the reviewer’s insightful comment. Indeed, the question of whether iCAD and ASCAD represent distinct immunological states or different stages of a shared disease continuum remains unresolved. Our cross-sectional design does not allow us to determine whether the observed immune differences reflect a temporal evolution or pre-existing differences in immune susceptibility. While our findings point toward divergent immune phenotypes, we agree that longitudinal studies will be required to elucidate this. We have now included this important consideration in the revised “Limitations” section of the Discussion.

Comment 5: It was also very difficult to follow the structure and interpretation of the predictive model. The manuscript would benefit from a clearer explanation of how variable importance was assessed and how these results should be interpreted in the context of disease classification. Clarifying whether these values reflect contribution to model accuracy, effect size, or another metric would significantly aid comprehension.

Response 5: We thank the reviewer for its valuable input. We have now added the requested information to the predictive model in the materials and methods section.

Importance function estimates the contribution of each predictor to the model’s performance through two metrics: Mean Decrease in Accuracy (MDA) and Mean Decrease in Gini (MDG). In this study, we report the MDA values, which are calculated by randomly permuting each variable and measuring the corresponding drop in classification accuracy on the out-of-bag (OOB) data. A larger decrease indicates a stronger impact of the variable on the model’s predictive performance. Thus, these scores reflect the relative importance of each variable in maintaining model accuracy, rather than direct effect sizes or statistical significance. This approach is particularly well suited for complex, non-linear interactions often found in immune and clinical datasets.

Comment 6: Additionally, a final graphical summary would help readers track the immune phenotypes associated with CAD, iCAD, and ASCAD.

Response 6: We thank the reviewer for the suggestion. We have created a summary figure of the results that we hope will help bringing more clarity to our study.

Reviewer 3 Report

Comments and Suggestions for Authors

Thank you for the opportunity to review your manuscript. I think that your paper is fascinating and well done. Your scientific work highlights a very important topic on the impact of inflammation on coronary and aortic valve diseases. I would just advise you to also cite in page 2, lines 72-73: Zanella et al. The Aortic Inflammation Affects Long-Term Freedom From Reintervention After Bentall Procedure. Circulation, 2024, 150.Suppl_1: A4147337-A4147337. This is a recent scientific work that fits well with what you said about inflammation and its impact on atherosclerotic-based diseases.

Author Response

Comment 1: Thank you for the opportunity to review your manuscript. I think that your paper is fascinating and well done. Your scientific work highlights a very important topic on the impact of inflammation on coronary and aortic valve diseases. I would just advise you to also cite in page 2, lines 72-73: Zanella et al. The Aortic Inflammation Affects Long-Term Freedom From Reintervention After Bentall Procedure. Circulation, 2024, 150.Suppl_1: A4147337-A4147337. This is a recent scientific work that fits well with what you said about inflammation and its impact on atherosclerotic-based diseases.

Response 1: We are very grateful for the reviewer’s kind words and for the thoughtful suggestion to include the citation: “Zanella et al. Circulation. 2024;150(Suppl_1): A4147337”.

We have carefully reviewed this abstract and fully agree that inflammation plays a pivotal role in both coronary and aortic valve pathophysiology, as also highlighted in our study. However, with respect, we feel that the specific context of the cited abstract—namely, long-term outcomes following the Bentall procedure—differs in scope from our current work, which focuses on immune dysregulation and senescence in peripheral blood profiles of CAD and ASCAD patients using flow cytometry. Additionally, the reference is currently only available as a conference abstract, which limits its utility as a peer-reviewed scientific source in the context of a primary research manuscript.

Therefore, while we greatly appreciate the reviewer’s insight, we respectfully believe that incorporating this particular citation may not substantially enhance the clarity or scientific foundation of the current manuscript. Should more detailed or peer-reviewed data from this study become available in the future, we would be pleased to consider it in subsequent work.

Reviewer 4 Report

Comments and Suggestions for Authors

Peer Review of "Divergent Immune Pathways in Coronary Artery Disease and Aortic Stenosis: The Role of Chronic Inflammation and Senescence"

Introduction and Purpose

The study addresses the underexplored intersection between CAD and aortic stenosis, examining immune cell dynamics and senescence markers. It tackles a significant health issue, Coronary Artery Disease (CAD), and its association with immune dysregulation, a novel and relevant area of research.

The paper aims to differentiate between isolated CAD (iCAD) and CAD with Aortic Stenosis (ASCAD) by investigating immune pathways, which are crucial for understanding disease mechanisms and developing targeted therapies.

Methodology

The use of flow cytometry to analyze immune populations in peripheral blood is appropriate and provides detailed insights into immune cell dynamics. The methodology is generally sound, including detailed flow cytometry protocols, well-characterized participant cohorts, and relevant statistical methods. However, a few concerns need to be addressed:

The manuscript mentions that the control group had equivalent epidemiological characteristics, which is not well-substantiated with data. More clarity is needed on matching for age, sex, and comorbidities.

For machine learning, validation involves using random forest models. However, I believe it is necessary to provide details on cross-validation (e.g., k-fold, leave-one-out) to avoid overfitting and ensure models generalize well to new data.

Results and Discussion

The findings reveal distinct immune dysregulation patterns in iCAD and ASCAD, with iCAD characterized by immune activation and ASCAD by immunosenescence and cytotoxicity. The study highlights the role of specific cytokines, such as IL-15 and fractalkine, in the pathogenesis of CAD, providing potential targets for therapeutic intervention. Is there any medication for this?

What are the clinical benefits of these differences? Can they be used as a therapeutic target or The relatively small sample size may limit the generalizability of the findings, and future studies should aim to include more diverse populations.

The study could benefit from longitudinal data to assess changes in immune profiles over time and their impact on disease progression.

Conclusion

The study concludes that iCAD is driven by immune activation, while ASCAD is characterized by immunosenescence and cytotoxicity. This suggests that ASCAD is not merely a more severe form of CAD but a distinct immune-mediated disease.

A machine learning model was developed to distinguish between iCAD and ASCAD accurately. The model identified CD4+ T-cell memory subsets and CD57 expression as key discriminators between the two conditions.

From a pathophysiological perspective, it brings new information—coronary artery disease has an inflammatory component (immune activation), while aortic stenosis has a degenerative one (immunosenescence and cytotoxicity)—which are known factors, without a clear understanding of the mechanisms and types of immune cells involved. This article sheds some light on this issue. Can these differences be used for risk stratification or become therapeutic targets?

Author Response

Comment 1: The manuscript mentions that the control group had equivalent epidemiological characteristics, which are not well-substantiated with data. More clarity is needed on matching for age, sex, and comorbidities.

Response 1: We thank the reviewer for its valuable feedback. We have clarified the characteristics of the control cohort in the materials and methods section.

Comment 2: For machine learning, validation involves using random forest models. However, I believe it is necessary to provide details on cross-validation (e.g., k-fold, leave-one-out) to avoid overfitting and ensure models generalize well to new data.

Response 2: We thank the reviewer for this important observation regarding model validation. Due to the limited sample size, especially in the disease subgroups, we opted to use the out-of-bag (OOB) error estimation intrinsic to the Random Forest algorithm as an internal cross-validation strategy. OOB estimates provide a reliable assessment of model generalizability in small datasets without the need for further data splitting, which could compromise model stability. This explanation has now been added to the “Materials and Methods” section to clarify our validation approach.

Comment 3: The study highlights the role of specific cytokines, such as IL-15 and fractalkine, in the pathogenesis of CAD, providing potential targets for therapeutic intervention. Is there any medication for this?

Response 3: We thank the reviewer for raising this relevant question. IL-15 and fractalkine (CX3CL1) are indeed emerging targets for immune modulation. Several IL-15–based immunotherapies, including superagonists (ALT-803, NKTR-255) and receptor antagonists, are currently in clinical trials, primarily for cancer and autoimmune diseases. Fractalkine inhibitors such as KAND567 and monoclonal antibodies like E6011 have shown promise in early trials for inflammatory and fibrotic conditions. However, to date, no clinical studies have evaluated these agents in the context of coronary artery disease (CAD). We have now briefly discussed these therapeutic avenues in the revised Discussion section of the manuscript.

Comment 4: What are the clinical benefits of these differences? Can they be used as a therapeutic target or the relatively small sample size may limit the generalizability of the findings, and future studies should aim to include more diverse populations.

Response 4: We appreciate the reviewer’s thoughtful comment. The immune phenotypes identified in our study, particularly those distinguishing iCAD and ASCAD, hold potential for both risk stratification and therapeutic targeting. However, as correctly noted, the relatively small and homogeneous sample size limits the generalizability of our findings. While our results suggest promising avenues, including the use of IL-15 and fractalkine-related pathways as targets, future studies are needed to validate these biomarkers in larger and more diverse cohorts, including patients at earlier stages of disease. We have now addressed these points in the revised Discussion section under “Limitations” and outlined the need for translational research to define clinical applications.

Comment 5: The study could benefit from longitudinal data to assess changes in immune profiles over time and their impact on disease progression.

Response 5: We thank the reviewer for its valuable input. Unfortunately, this was a three-year project with limited budget and no longitudinal follow up was possible. We plan to extend this study in the near future. This limitation has been addressed in the discussion section.

Comment 6: From a pathophysiological perspective, it brings new information—coronary artery disease has an inflammatory component (immune activation), while aortic stenosis has a degenerative one (immunosenescence and cytotoxicity)—which are known factors, without a clear understanding of the mechanisms and types of immune cells involved. This article sheds some light on this issue. Can these differences be used for risk stratification or become therapeutic targets?

Response 6: We thank the reviewer for this important reflection. Indeed, one of the main contributions of our study is the identification of divergent immune mechanisms in iCAD and ASCAD—immune activation versus immunosenescence—which may underlie distinct clinical trajectories. These findings offer a pathophysiological rationale for using immune profiles as potential tools for risk stratification. Moreover, as discussed, several of the implicated pathways, including IL-15 signalling and CX3CR1-mediated migration, may represent targets for therapeutic intervention. While further validation in larger cohorts is needed, our results support the potential clinical utility of immune profiling in CAD management. Early diagnosis and proper treatment may have a major impact in disease progression.

Round 2

Reviewer 1 Report

Comments and Suggestions for Authors

I believe that all my comments have been properly addressed and that the manuscript can be published in its current form.

Reviewer 2 Report

Comments and Suggestions for Authors

 The authors have satisfactorily addressed the raised comments and questions, significantly enhancing the understanding of the manuscript's initial limitations and inquiries.